

# Automatic modular design of robot swarms using behavior trees as a control architecture

Antoine Ligot[1,*], Jonas Kuckling[1,*], Darko Bozhinoski[1,2] and Mauro Birattari[1]

[1] IRIDIA, Université Libre de Bruxelles, Brussels, Belgium
[2] Cognitive Robotics, Delft University of Technology, Delft, Netherlands
* These authors contributed equally to this work.

## ABSTRACT

We investigate the possibilities, challenges, and limitations that arise from the use of behavior trees in the context of the automatic modular design of collective behaviors in swarm robotics. To do so, we introduce `Maple`, an automatic design method that combines predefined modules—low-level behaviors and conditions—into a behavior tree that encodes the individual behavior of each robot of the swarm. We present three empirical studies based on two missions: AGGREGATION and FORAGING. To explore the strengths and weaknesses of adopting behavior trees as a control architecture, we compare `Maple` with `Chocolate`, a previously proposed automatic design method that uses probabilistic finite state machines instead. In the first study, we assess `Maple`'s ability to produce control software that crosses the reality gap satisfactorily. In the second study, we investigate `Maple`'s performance as a function of the design budget, that is, the maximum number of simulation runs that the design process is allowed to perform. In the third study, we explore a number of possible variants of `Maple` that differ in the constraints imposed on the structure of the behavior trees generated. The results of the three studies indicate that, in the context of swarm robotics, behavior trees might be appealing but in many settings do not produce better solutions than finite state machines.

# INTRODUCTION

In this article, we extend the original definition of AutoMoDe—the family of automatic modular design methods proposed by *Francesca et al. (2014)*—to study the use of behavior trees as a control software architecture for robot swarms.

In swarm robotics, a large group of autonomous robots cooperate to perform a mission that is beyond the limited capabilities of a single robot (*Beni, 2004*; *Şahin, 2004*; *Brambilla et al., 2013*; *Garattoni & Birattari, 2016*). A robot swarm is highly redundant, self-organized, and decentralized in nature. These properties are appealing in applications that, for example, imply a high risk of individual failure, take place in locations with limited communication infrastructure, or require scalability (*Dorigo, Birattari & Brambilla, 2014*).

Corresponding authors
Antoine Ligot, aligot@ulb.ac.be
Mauro Birattari, mbiro@ulb.ac.be

Unfortunately, these properties have also a downside: it is difficult to conceive and implement control software for the individual robots so that a desired collective behavior is produced. As a general methodology is still missing, the design process is typically labor intensive, time consuming, error prone, and difficult to reproduce (*Brambilla et al., 2013*; *Francesca & Birattari, 2016*; *Bozhinoski & Birattari, 2018*).

Automatic design is a valid and promising alternative (*Francesca & Birattari, 2016*; *Birattari et al., 2019*). In automatic design, the problem of designing control software to perform a given mission is re-formulated into an optimization problem: an optimization algorithm searches a space of candidate solutions so as to maximize an objective function. In this context, a candidate solution is an instance of the control software to be executed by each robot; and the objective function is a mission-dependent score that measures the performance of the swarm on the given mission. Because the evaluation of candidate solutions on physical robots is costly and time consuming, automatic design methods typically rely on simulation.[1] A major issue with the adoption of simulation in automatic design is the so called *reality gap* (*Brooks, 1992*; *Jakobi, Husbands & Harvey, 1995*): the difference between simulation and reality, which is ultimately unavoidable. As a result of the reality gap, it is possible, and even likely, that control software generated in simulation suffers from a drop in performance when deployed in reality. The reality gap is one of the most challenging issues in the automatic design of robot swarms (*Francesca & Birattari, 2016*).

Evolutionary swarm robotics (*Trianni, 2008*, *2014*)—the application of evolutionary robotics (*Lipson, 2005*; *Floreano, Husbands & Nolfi, 2008*) to robot swarms—is a popular automatic design approach. In evolutionary swarm robotics, an evolutionary algorithm (*Bäck, Fogel & Michalewicz, 1997*) generates the control software of the robots, typically in the form of an artificial neural network. The input of the artificial neural network are the sensor readings; the output are the control actions that drive the actuators. Although evolutionary swarm robotics has been successfully used to generate control software for various missions (*Quinn et al., 2003*; *Christensen & Dorigo, 2006*; *Hauert, Zufferey & Floreano, 2009*; *Trianni & Nolfi, 2009*), it presents some known limitations, among which is its inability to cross the reality gap reliably (*Silva et al., 2016*). *Francesca et al. (2014)* conjectured that the issues encountered by evolutionary robotics with the reality gap are due to the high representational power of artificial neural networks. This leads the design process to overfit characteristics of the simulator that do not have a counterpart in reality. As a result, the control software produced fails to generalize to the real world.

Inspired by the notion of *bias–variance tradeoff* (*Geman, Bienenstock & Doursat, 1992*) from the supervised learning literature, *Francesca et al. (2014)* developed AutoMoDe: an automatic design approach in which control software is conceived by automatically assembling predefined modules (that is, low-level behaviors and conditions) into a modular software architecture. The rationale behind AutoMoDe is to lower the representational power—and therefore the variance—of the control software it produces by introducing bias: it is restricted to be combinations of predefined modules. This restriction restrains the space of the possible instances of control software that can be

[1] For the sake of completeness, we mention here that some automatic design methods do not rely on simulation. They operate while robots are deployed in their operating environment. We refer the reader to *Francesca & Birattari (2016)* for a discussion of advantages and limitations of these methods.

generated by AutoMoDe, with the intent of reducing the risk to overfit characteristics of the simulator that are not a faithful representation of reality.

AutoMoDe is an abstract approach that, in order to be used, must be specialized to a specific robotic platform by defining low-level behaviors and conditions, the specific rules/constraints to combine them, and the optimization algorithm to search the space of solutions. So far, all instances of AutoMoDe—that is, `Vanilla` (*Francesca et al., 2014*), `Chocolate` (*Francesca et al., 2015*), `Gianduja` (*Hasselmann, Robert & Birattari, 2018*), `Waffle` (*Salman, Ligot & Birattari, 2019*), `Coconut` (*Spaey et al., 2019*), `Icepop` (*Kuckling, Ubeda Arriaza & Birattari, 2019*), and `TuttiFrutti` (*Garzón Ramos & Birattari, 2020*)—that have been proposed target the e-puck robot (*Mondada et al., 2009*). To substantiate their conjecture, *Francesca et al. (2014)* compared the performance of `Vanilla` and `Chocolate` with `EvoStick`, an implementation of the classical evolutionary swarm robotics approach. In their experiments, *Francesca et al. (2014, 2015)* observed that both `Vanilla` and `Chocolate` are able to generate control software that crosses the reality gap satisfactorily. In addition, they observed what can be called a *rank inversion* (*Ligot & Birattari, 2018, 2019*): `EvoStick` outperforms `Vanilla` and `Chocolate` in simulation, but `Vanilla` and `Chocolate` outperform `EvoStick` in reality.

In the original definition, *Francesca et al. (2014)* have characterized AutoMoDe as an approach to generate control software in the form of a probabilistic finite-state machine (*Francesca et al., 2014, 2015*). However, this characterization appears to be too restrictive: the element that truly characterize AutoMoDe—whose name is the contraction of *automatic modular design*—is that it generates control software by combining and fine-tuning predefined modules. Indeed, according to the conjecture of *Francesca et al. (2014)*, its modular nature is the main reason why AutoMoDe has shown to be robust to the reality gap: the architecture into which the modules are assembled appears to be a secondary issue.

In this article, we aim at investigating the possibilities, challenges, and limitations that arise from the use of behavior trees in the context of the automatic modular design of collective behaviors in swarm robotics. Behavior trees are a popular control architecture originally proposed for game development (*Champandard, 2007*; *Champandard, Dawe & Hernandez-Cerpa, 2010*), and offer a number of advantages over finite-state machines, such as enhanced expressiveness, inherent modularity, and two-way control transfers (*Colledanchise & Ögren, 2018*). Moreover, *Colledanchise & Ögren (2018)* have shown that behavior trees generalize a number of other architectures including the subsumption architecture (*Brooks, 1986*) and decision trees (*Nehaniv & Dautenhahn, 2002*). Recently, behavior trees have attracted interest from the domains of artificial intelligence and robotics (*Colledanchise & Ögren, 2018*).

The main characteristics of behavior trees is the use of complex behavioral modules as leaf nodes that return their state of execution: *running*, *success*, or *failure*. Behavior trees are therefore a convenient way to implicitly model plans of execution: they define what action needs to be taken if a given condition is met or not, and if a given behavior succeeds or fails. The current practice of swarm robotics goes against the principle of planning as the individual robots used are typically simple and reactive in the sense defined by

*Brooks (1991)*. In the reactive paradigm, a low-level behavior is executed indefinitely until an external event triggers the indefinite execution of another low-level behaviors, and so on. Due to this cultural legacy, the low-level behaviors typically operated by robots within swarms do not have natural termination criteria, and therefore do not have success/failure states. In addition, the hardware limitations of the simple individual robots typically used in swarm robotics does not give them the capabilities of assessing natural termination criteria of the low-level behaviors they are executing. It is nonetheless possible to use behavior trees as a control software architecture for robot swarms, as it has already been done by *Jones et al. (2018)*. However, to do so, design choices are needed and possibly only a subset of the functionalities of the behavior trees can be used, which forces one to renounce the implicit planning that they offer. For example, *Jones et al. (2018)* considered atomic commands as action nodes (i.e., move forward, turn left/right, or store data) that always return success after the second execution of the behavior, and never return failure. Despite not benefiting from the full potential of behaviors trees when combining low-level behaviors without natural termination criteria, it remains that the inherent modularity that they offer makes behavior trees a control architecture that is well worth exploring in the context of automatic design of robot swarm. Indeed, because each subtree is a valid structure, behavior trees are more easily manipulated than finite-state machines (*Colledanchise & Ögren, 2018*). Therefore, one could conceive tailored optimization algorithms based on local manipulations that explore the possible collective behaviors obtained by selecting, combining, and fine-tuning predefined modules into behavior trees more efficiently than into finite state machines.

In this work, we study the use of behavior trees in fully automatic off-line design of robot swarms (*Birattari, Ligot & Hasselmann, 2020*). We do so by developing a method that uses low-level behaviors that are more complex than those of *Jones et al. (2018)*, but yet less complex than those typically used in applications of behaviors tree to other domains. Indeed, rather than using atomic commands and assuming the artificial return of success after a given time, we use low-level behaviors as they are typically conceived in swarm robotics, that is, without the notion of success or failure. We devised `Maple`, a novel instance of AutoMoDe that has at its disposal the same low-level behaviors and conditions used by `Vanilla` and `Chocolate`, with the goal of understanding the conditions under which it is beneficial to adopt behavior trees over finite state machines in modular automatic design. `Maple` is in many aspects similar to `Chocolate`: in fact, we only substituted probabilistic finite state machines with behavior trees. This way, differences in performance between the two methods can only be attributed to the different control architecture they adopt. Because the behavioral modules adopted in `Maple` only return *running*, `Maple` produces control software in the form of behavior trees with predetermined structure that only use a subset of the behavior trees functionalities. In this structure, a conditional module is combined with a low-level behavior in order to act as a termination criterion for the said low-level behavior. We present three empirical studies conducted on two missions. In the first one, we study the robustness of automatically generated control software in the form of behavior trees by comparing its performance in simulation and in reality. The results show that the control software generated by `Maple`

performs similarly to the one generated by `Chocolate`, and that it crosses the reality gap more satisfactorily than the one generated by `EvoStick`. This confirms *Francesca et al.'s (2014)* conjecture that AutoMoDe is robust to the reality gap due to its modular nature. In the second study, we investigate the impact of different design budgets on the performance of the control software produced by `Maple` and `Chocolate`. The results indicate that `Maple` converges to satisfactory solutions faster than `Chocolate`. However, it appears that the expressiveness of the control structure adopted in `Maple` is reduced with regard to the one of finite state machines: `Maple` cannot produce solutions as complex as those produced by `Chocolate` for one of the missions considered. In the third study, we explore multiple alternatives to the structure of the behavior trees adopted in `Maple`. All these alternatives are predefined, restricted behavior trees structures that can be used with low-level behaviors that do not have a natural termination criterion. The results show that none of the explored structures outperform the one adopted in `Maple` in both missions considered. This paper extends on preliminary results presented in a conference (*Kuckling et al., 2018a*). We present here the complete description of the automatic design method and justifications of the design choices, together with more experimental results.

The work of *Jones et al. (2018)* brought initial evidence that behavior trees are a viable control architecture to be adopted in swarm robotics when considering atomic commands as action nodes. Our studies highlight the strengths and weaknesses of behavior trees when applied low-level behaviors as they are typically conceived in this domain: our results suggest that, although behavior trees might be appealing under some settings, under other they do not produce better results than finite state machines and might be even outperformed by the latter. What hinders the application of behavior trees to swarm robotics is the absence of the notion of success and failure in the low-level behaviors typically adopted in swarm robotics. We believe that, in order to develop low-level behaviors that are appropriate for behavior trees, one should overcome technical issues (that is, use robots whose hardware capabilities enable them to infer natural termination criteria) and a cultural legacy.

## Behavior trees

In this section, we give a brief description of behavior trees and their functioning. We adopt the framework that *Marzinotto et al. (2014)* proposed to unify the different variants of behavior trees described in the literature. We refer the reader to the original description of the framework for more details.

The original idea of behavior trees was proposed for the Halo 2 video game (*Isla, 2005*). Since then, behavior trees have found applications in many computer games, for example, Spore and Bioshock (*Champandard, Dawe & Hernandez-Cerpa, 2010*). Recently, behavior trees have attracted the interest of the research community. Initial research focused on the automatic generation of behaviors in video games, for example, the commercial game DEFCON (*Lim, Baumgarten & Colton, 2010*) and the Mario AI competition (*Perez et al., 2011*). Even more recently, behavior trees have found applications in the control of unmanned aerial vehicles (*Ögren, 2012*), surgical robots (*Hu et al., 2015*), and collaborative robots (*Paxton et al., 2017*).

A behavior tree is a control architecture that can be expressed as a directed acyclic graph with a single root. With a fixed frequency, the root generates a *tick* that controls the execution. The tick is propagated through the tree and activates each node that it visits. The path that the tick takes through the tree is determined by the inner nodes, which are called control-flow nodes. Once the tick reaches a leaf node, a condition is evaluated or an action is performed. Then, the leaf node immediately returns the tick to its parent together with one of the following three values: *success*, *failure*, or *running*. A condition node returns *success*, if its associated condition is fulfilled; *failure*, otherwise. An action node performs a single control step of its associated action and returns *success*, if the action is completed; *failure*, if the action failed; *running*, if the action is still in progress. When a control-flow node receives a return value from a child, it either immediately returns this value to its parent, or it continues propagating the tick to the remaining children. There are six types of control-flow nodes:

**Sequence** ($\rightarrow$): ticks its children sequentially, starting from the leftmost child, as long as they return *success*. Because it does not remember the last child that returned *running*, it is said to be memory-less. Once a child returns *running* or *failure*, the sequence node immediately passes the returned value, together with the tick, to its parent. If all children return *success*, the node also returns *success*.

**Selector** (?): memory-less node that ticks its children sequentially, starting from the leftmost child, as long as they return *failure*. Once a child returns *running* or *success*, the selector node immediately passes the returned value, together with the tick, to its parent. If all children return *failure*, the node also returns *failure*.

**Sequence*** ($\rightarrow^*$): version of the sequence node with memory: resumes ticking from the last child that returned *running*, if any.

**Selector*** (?*): version of the selector node with memory: resumes ticking from the last child that returned *running*, if any.

**Parallel** ($\Rightarrow$): ticks all its children simultaneously. It returns *success* if a defined fraction of its children return *success*; *failure* if the fraction of children return *failure*; *running* otherwise.

**Decorator** ($\delta$): is limited to a single child. It can alter the number of ticks passed to the child and the return value according to a custom function defined at design time.

In the context of automatic modular design, the most important properties of behavior trees are their enhanced expressiveness, the principle of two-way control transfers, and their inherent modularity (*Ögren, 2012*; *Colledanchise & Ögren, 2018*). Ögren and coworkers have shown that behavior trees generalize finite-state machines only with selector and sequence nodes (*Ögren, 2012*; *Marzinotto et al., 2014*). With parallel nodes, behavior trees are able to express individual behaviors that have no representation in classical finite-state machines. The principle of two-way control transfers implies that the control can be passed from a node to its child, and can also be returned from the child, along with information about the state of the system. Finally, behavior trees are inherently modular: each subtree is a valid behavior tree. Due to this property, behavior trees can be easily manipulated as one can move, modify, or prune subtrees without compromising the structural integrity of the behavior tree. The modularity of behavior trees could simplify the conception of tailored optimization algorithm based on local manipulations.

**Table 1 Reference model RM1.1 (*Hasselmann et al., 2018*). Sensors and actuators of the e-puck robot. Period of control cycle: 100 ms.**

| Sensor/Actuator | Variables | Values |
| --- | --- | --- |
| Proximity | $prox_i$, with $i \in \{0,\ldots,7\}$ | [0,1] |
| Light | $light_i$, with $i \in \{0,\ldots,7\}$ | [0,1] |
| Ground | $ground_i$, with $i \in \{0,\ldots,2\}$ | {*black, gray, white*} |
| Range-and-bearing | $n$ | $\{0,\ldots,19\}$ |
|  | $V_d$ | ([0,0.7]m, [0,2$\pi$] radian) |
| Wheels | $v_l, v_r$ | $[-0.12, 0.12]\text{ms}^{-1}$ |

## AUTOMODE—MAPLE

`Maple` is an automatic modular design method that generates control software in the form of behavior trees. It does so by selecting, combining, and fine-tuning a set of predefined modules: the six low-level behaviors and the six conditions defined by *Francesca et al. (2014)* for `Vanilla`, and later used in `Chocolate` (*Francesca et al., 2015*). We introduce `Maple` with the purpose of exploring the use of behavior trees as a control architecture in the automatic modular design of robot swarms. To conduct a meaningful study on the potentials of behavior trees as a control architecture, we compare `Maple` with `Chocolate`, a state-of-the-art automatic modular design method that generates control software in the form of probabilistic finite-state machines (*Francesca et al., 2015*; *Francesca & Birattari, 2016*). We conceived `Maple` to be as similar as possible to `Chocolate` so that differences in performance between the two methods can only be attributed to the different control architecture they adopt. `Maple` and `Chocolate` generate control software for the same robotic platform, they have at their disposal the same set of modules, and they use the same optimization algorithm.

In a probabilistic finite-state machine generated by `Chocolate`, a state is an instantiation of a low-level behavior and a transition is an instantiation of a condition. Because low-level behaviors (the states of the finite-state machine) are executed until an external condition (a transition) is enabled, they do not have inherent termination criteria. The absence of natural termination criteria implies that, when used as action nodes in a behavior tree generated by `Maple`, the low-level behaviors of `Chocolate` can only return *running*. As a result, part of the control-flow nodes of behavior trees do not work as intended. With `Maple`, we chose to use the unmodified modules of `Chocolate`, and force the generated behavior trees to adopt a restricted structure that only uses a subset of the control-flow nodes.

### Robotic platform

`Maple` produces control software for the e-puck robot (*Mondada et al., 2009*) equipped with several extension boards (*Garattoni et al., 2015*), including the range-and-bearing board (*Gutiérrez et al., 2009*). The predefined modules on which `Maple` operates have access to a subset of the capabilities of the e-puck robot that are formally defined by the reference model RM 1.1 (*Hasselmann et al., 2018*)—see Table 1.

The modules can adjust the velocity of the two wheels ($v_l$ and $v_r$) of the robot, detect the presence of obstacles ($prox_i$), measure the intensity of the ambient light ($light_i$), and identify whether the ground situated directly beneath the robot is black, gray, or white ($ground_i$). The modules have also access to the number $n$ of surrounding peers within a range of up to 0.7 m, as well as to a vector $V_d = \sum_{m=1}^{n}(1/r_m, \angle b_m)$ where $r_m$ and are distance and bearing of the $m$-th neighboring peer (*Spears et al., 2004*).

## Set of modules

`Maple` has at its disposal the same set of modules used by `Vanilla` (*Francesca et al., 2014*) and `Chocolate` (*Francesca et al., 2015*). Some of the modules are parametric so that the optimization algorithm can fine-tune their behavior on a per-mission basis. The set comprises six low-level behaviors and six conditions. A low-level behavior is a way in which the robot operates its actuators in response to the readings of its sensors. A condition is a context that the robot perceives via its sensors. Conditions contribute to determine which behavior is executed at any moment in time.

In the behavior trees generated by `Maple`, an action node is selected among the six low-level behaviors and a condition node is selected among the six conditions. In the following, we briefly describe the low-level behaviors and conditions. For the details, we refer the reader to their original description given by *Francesca et al. (2014)*.

### *Low-level behaviors*

**Exploration:** if the front of the robot is clear of obstacles, the robot moves straight. When an obstacle is perceived via the front proximity sensors, the robot turns in-place for a random number of control cycles drawn in $\{0, \ldots, \tau\}$. $\tau$ is an integer parameter $\in \{0, \ldots, 100\}$.

**Stop:** the robot does not move.

**Phototaxis:** the robot moves towards the light source. If no light source is perceived, the robot moves straight while avoiding obstacles.

**Anti-phototaxis:** the robot moves away from the light source. If no light source is perceived, the robot moves straight while avoiding obstacles.

**Attraction:** the robot moves towards its neighboring peers, following $\alpha V_d$, where the parameter $\alpha \in [1, 5]$ controls the speed of convergence towards them. If no peer is perceived, the robot moves straight while avoiding obstacles.

**Repulsion:** the robot moves away from its neighboring peers, following $-\alpha V_d$, where the parameter $\alpha \in [1, 5]$ controls the speed of divergence. If no peer is perceived, the robot moves straight while avoiding obstacles.

### *Conditions*

**Black-floor:** true with probability $\beta$, if the ground situated below the robot is perceived as black.

**Gray-floor:** true with probability $\beta$, if the ground situated below the robot is perceived as gray.

**White-floor:** true with probability $\beta$, if the ground situated below the robot is perceived as white.

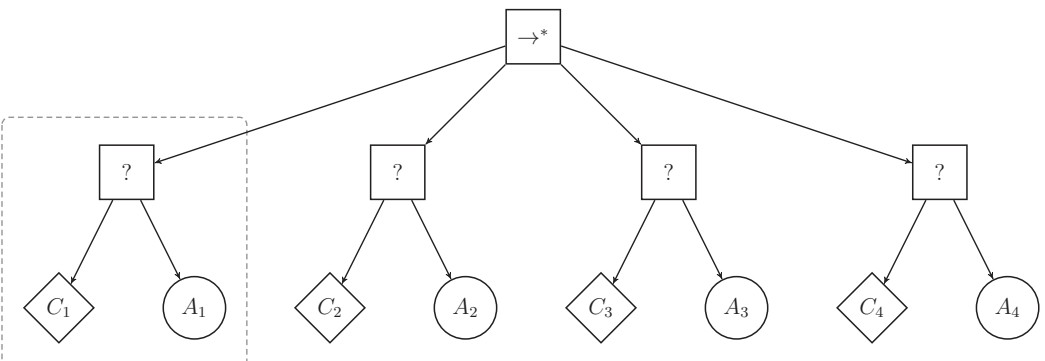

**Figure 1 Illustration of a behavior tree with restricted structure that** `Maple` **can produce.** `Maple` generates a behavior tree by defining first the number of selector subtrees (highlighted by the dashed box), and by then specifying and fine-tuning the condition and action nodes that compose each selector subtree.

**Neighbor-count:** true with probability $z(n) = \left(1 + e^{\eta(\xi-n)}\right)^{-1}$, where $n$ is number of detected peers. The parameters $\eta \in [0, 20]$ and $\xi \in \{0, \dots, 10\}$ control the steepness and the inflection point of the function, respectively.

**Inverted-neighbor-count:** true with probability $1 - z(n)$.

**Fixed-probability:** true with probability $\beta$.

## Control software architecture

The low-level behaviors of `Chocolate` have no inherent success or failure criterion and can only return *running* when used as action nodes in behavior trees. To use `Chocolate`'s low-level behaviors as action nodes, we constrained `Maple` to generate behavior trees that have a particular, restricted structure. This restricted structure only uses a subset of the control-flow nodes of the classical implementation of behavior trees. The top-level node is a sequence* node ($\rightarrow^*$) and can have up to four selector subtrees attached to it. A selector subtree is composed of a selector node (?) with two leaf nodes: a condition node as the left leaf node, and an action node as the right leaf node. Figure 1 illustrates a behavior tree with the restricted structure adopted here. We limit the maximal number of subtrees, and therefore the number of action nodes, to four so as to mimic the restrictions of `Chocolate`, which generates probabilistic finite-state machines with up to four states.

In the example of Fig. 1, the left-most selector subtree (highlighted by the dashed box) is first ticked and action $A_1$ is executed as long as condition $C_1$ returns *failure*. If condition $C_1$ returns *success*, the top-level node ($\rightarrow^*$) ticks the second selector subtree, and $A_2$ is executed, provided that $C_2$ returns *failure*. Because the top-level node is a control-flow node with memory, the tick will resume at the second subtree in the following control cycle. $A_2$ is therefore executed as long as $C_2$ returns *failure*. Although actions $A_1$ and $A_4$ are not in adjacent sub-trees, $A_4$ can be executed directly after $A_1$ granted that conditions $C_1$, $C_2$, and $C_3$ return *success* and $C_4$ returns *failure*. When condition $C_4$ of the last selector subtree returns *success*, the top-level node of the tree also returns *success* and no

action is performed. In this case, the tree is ticked again at the next control cycle, and the top-level node ticks the left-most selector subtree again.

The size of the space spanning all possible instance of control software that can be produced by `Maple` is in $O(|\mathscr{B}|^4|\mathscr{C}|^4)$, where $\mathscr{B}$ and $\mathscr{C}$ are the sets of low-level behaviors and conditions, respectively (*Kuckling et al., 2018b*). The search space can be formally defined as:

$$\left[T, \#N^{(2)}, N_i^{(2)}, \#L_i, L_{ij}, L_{ij}^p\right], \quad with\ i = \left\{1, ..., \#N^{(2)}\right\}, j = \{1, ..., \#L_i\},$$

where $T \in \{\text{sequence}^*\}$ is the type of the top-level node; $\#N^{(2)} \in \{1, ..., 4\}$ is the number of level 2 nodes; $N_i^{(2)} \in \{\text{selector}\}$ is the type of the level 2 node $i$; $\#L_i \in \{2\}$ is the number of leafs of node $i$; $L_{ij}$ is the type of the $j$-th leaf of node $i$, with $L_{i1} \in \mathscr{C}$ and $L_{i2} \in \mathscr{B}$; and $L_{ij}^p$ are the parameters of leaf $L_{ij}$.

### Optimization algorithm

`Maple` uses Iterated F-race (*Balaprakash, Birattari & Stützle, 2007*; *López-Ibáñez et al., 2016*) as an optimization algorithm. Iterated F-race searches the space of all possible candidate solutions for the best one according to a mission-specific measure of performance. The Iterated F-race algorithm comprises multiple steps, each of which is reminiscent of a race. In the first race, a uniformly distributed set of candidate solutions is sampled. These candidates are initially evaluated on a set of different instances. Typically, an instance describes the configuration of the arena at the beginning of an experiment (that is, positions and orientations of the robots, positions of eventual obstacles or objects of interest, or color of the floor). After the initial set of evaluations is performed, a Friedman test (*Friedman, 1937, 1939*; *Conover, 1999*) is performed on the performance obtained by the candidate solutions. The candidate solutions that perform significantly worse than at least another one are discarded. The algorithm keeps evaluating the remaining candidate solutions on new instances and discards those that are statistically dominated. The race terminates when only one surviving candidate solution remains, or when the maximal number of evaluation defined for the race is reached. In the following races, the new set of candidate solutions is sampled with a distribution that gives higher priority to solutions that are similar to the surviving solutions of the previous one.

## EXPERIMENTAL SETUP

In this section, we describe the experimental setup that is common to the three studies conducted. In particular, we describe the previously proposed automatic design methods against which we compare `Maple`, the missions for which we generate control software, and the protocol we follow. Further details are given in each of the sections dedicated to the specific studies.

### Automatic design methods

In Study 1 and 2, we compare `Maple` with two previously proposed methods: `Chocolate` and `EvoStick`. `Maple` is described in the previous section. Here, we briefly describe

`Chocolate` and `EvoStick`: we refer the reader to *Francesca et al. (2014, 2015)* for the details.

   `Chocolate` (*Francesca et al., 2015*) is an automatic modular method that selects, combines, and fine-tunes the same twelve predefined modules as `Maple`. In `Chocolate`, the architecture of the control software is a probabilistic finite-state machine. In this context, a state is an instance of low-level behavior and an edge is an instance of condition. Similarly to `Maple`, `Chocolate` adopts Iterated F-race as an optimization algorithm. With `Chocolate`, the search space of Iterated F-race is restricted to probabilistic finite-state machines that comprise up to four states, and up to four outgoing edges per state. The size of the search space defined by the control architecture of `Chocolate` is in $O(|\mathscr{B}|^4|\mathscr{C}|^{16})$, where $\mathscr{B}$ and $\mathscr{C}$ are the sets of low-level behaviors and of conditions, respectively (*Kuckling et al., 2018b*).

   `EvoStick` (*Francesca et al., 2014, 2015*) is a straightforward implementation of the evolutionary swarm robotics approach. In `EvoStick`, the architecture of the control software is a fully-connected, single layer, feedforward neural network. The neural network comprises 24 input nodes for the readings of the sensors described in the reference model RM 1.1: 8 for the proximity sensors, 8 for the light sensors, 3 for ground sensors, and 5 for the range-and-bearing board. Out of the five input nodes dedicated to the range-and-bearing board, one is allocated to the number of detected peers and the four others are allocated to the scalar projection of the vector $V_d$ on four unit vectors. The neural network comprises 2 output nodes controlling the velocities of the wheels. The topology of the neural network is fixed, and an evolutionary algorithm fine-tunes the 50 weights of the connections between the input and the output nodes. Each weight is a real value in the range $[-5, 5]$. In `EvoStick`, the population is composed of 100 individuals that are evaluated 10 times per generation.

## Missions

We consider two missions: FORAGING and AGGREGATION. The two missions must be performed in a dodecagonal arena delimited by walls and covering an area of 4.91. The swarm is composed of 20 e-puck robots that are distributed uniformly in the arena at the beginning of each experimental run, and we limit the duration of the missions to 120.

### FORAGING

Because the robots cannot physically carry objects, we consider an idealized form of foraging. In this version, we reckon that an item is picked up when a robot enters a source of food, and that a robot drops a carried item when it enters the nest. A robot can only carry one item at a time. In the arena, a source of food is represented by a black circle, and the nest is represented by the white area (see Fig. 2). The two black circles have a radius of 0.15, they are separated by a distance of 1.2, and are located at 0.45 from the white area. A light source is placed behind the white area to indicate the position of the nest to the robots.

   The goal of the swarm is to retrieve as many items as possible from the sources to the nest. In other words, the robots must go back and forth between the black circles and

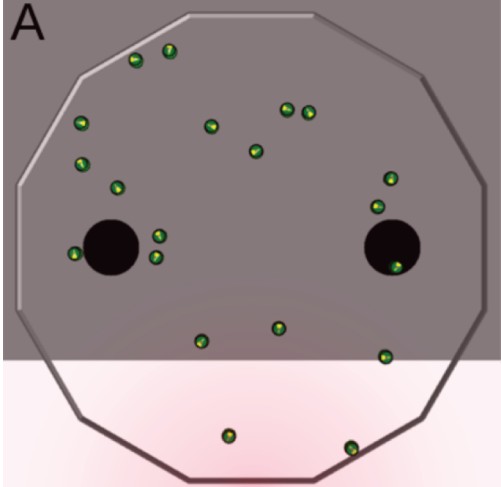 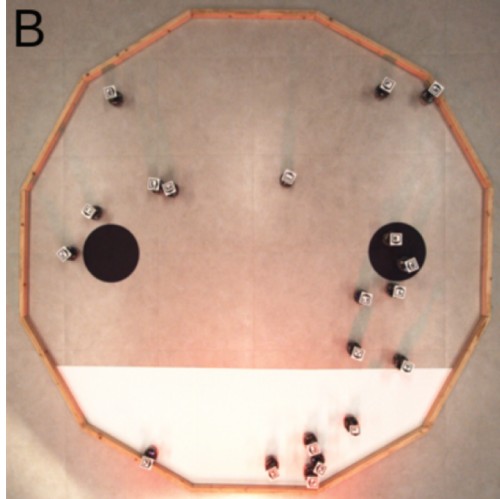

**Figure 2 FORAGING.** (A) Simulated arena. (B) Real arena. The red glow visible in the picture is due to a red gel we placed in front of the light source. With the red gel, the light does not disturb the overhead camera that is used to track the position of the robots and compute the objective function. Yet, the light is still perceived by the robots that use their infrared sensors to sense it.

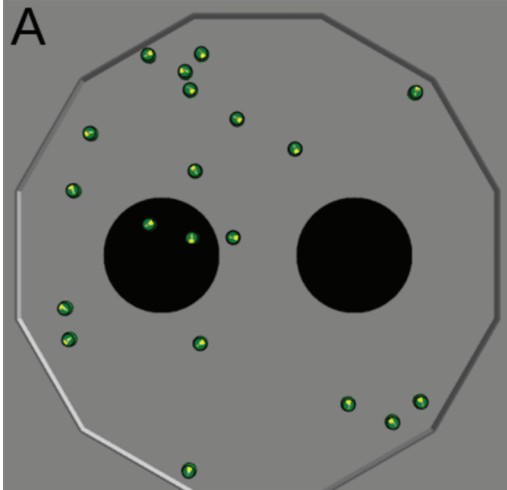 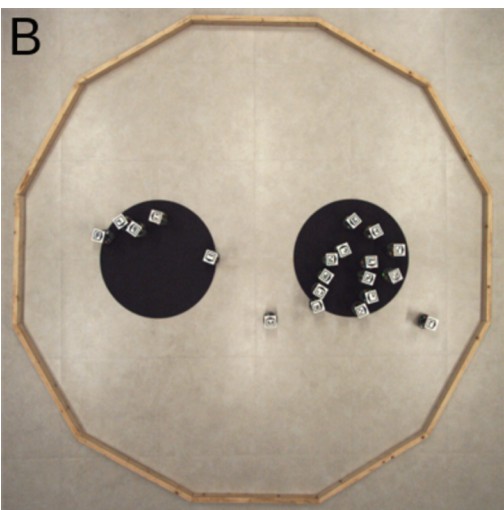

**Figure 3 AGGREGATION.** The objective function $F_A$ is computed as the maximal fraction of robots situated either on the left area ($N_l/N$) or on the right area ($N_r/N$). It is evaluated at the end of an experimental run. (A) Simulated arena, with $F_A = 0.1$ as 2 robots stand on the left black area ($N_l = 2$) and no robot stands on the right one ($N_r = 0$). (B) Real arena, with $F_A = 0.65$ as $N_l = 5$ and $N_r = 13$.

the white area as many times as possible. The objective function is $F_F = I$ where $I$ is the number of items deposited in the nest.

**AGGREGATION**

The swarm must select and aggregate in one of the two black areas (see Fig. 3). The two black areas have a radius of 0.3 and are separated by a distance of 0.4. The objective function is $F_A = max(N_l, N_r)/N$, where $N_l$ and $N_r$ are the number of robots located on the

left and right black area, respectively; and $N$ is the total number of robot in the swarm. The objective function is computed at the end of a run and is maximized when all the robots have aggregated in the same black area.

## Dummy control software

Throughout the three studies, we compare the performance of automatically generated control software to the one of two instances of control software—one per mission—that we call "dummy" control software. They perform a simple, naive, and trivial behavior that we can consider as a baseline for each mission. With this comparison, we assess whether the automatic design methods can produce behaviors that are more sophisticated than trivial solutions. To produce the two instances of dummy control software, we used the same low-level behaviors and conditions that `Maple` and `Chocolate` have at their disposal to generate control software. For FORAGING, we consider a strategy in which the robots move randomly in the environment. We obtained this strategy by using the low-level behavior *exploration*. For AGGREGATION, we consider a strategy in which the robots explore the environment randomly, and stop when they encounter a black spot. We obtained this strategy by combining the modules *exploration*, *black-floor*, and *stop*. To fine-tune the parameters of the modules, we used Iterated F-race with a design budget of 1k simulation runs.

## Methodology

To account for the stochasticity of the design process, we execute each design method several times, and therefore produce several instances of control software. The number of executions of the design methods varies with the study. To evaluate the performance of a design method, each instance of control software is executed once in simulation. In Study 1, each instance of control software is also executed once in reality.

Simulations are performed with ARGoS3, version beta 48 (*Pinciroli et al., 2012*; *Garattoni et al., 2015*). In the experiments with the robots, we use a tracking system comprising an overhead camera and QR-code tags on the robots to identify and track them in real time (*Stranieri et al., 2013*). With this tracking system, we automatically measure the performance of the swarm, and we automatically guide the robots to the initial position and orientation for each evaluation run. During an evaluation run, the robots may tip over due to collisions. To avoid damages, we intervene to put them upright.

In the three studies, we present the performance of the design methods in the form of box-and-whiskers boxplots. In addition, we present the median performance of the dummy control software assessed in simulation with a dotted horizontal line. In each study, statements such as "method $A$ is significantly better/worse than $B$" imply that significance has been assessed via a Wilcoxon rank-sum test, with confidence of at least 95%. The instances of control software produced, the experimental data collected in simulation and in reality, and videos of the behavior displayed by the swarm of physical robots are available online as Supplemental Material (*Ligot et al., 2020*).

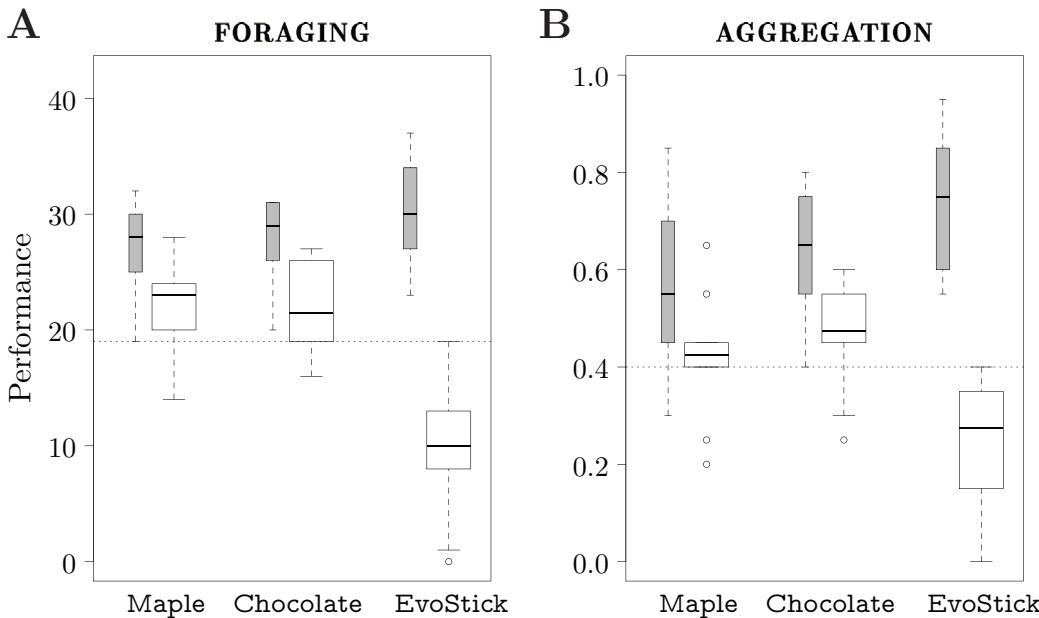

**Figure 4** **Results of Study 1.** The gray boxes represent the performance assessed in simulation; the white boxes the one assessed in reality. The dotted line represents the median performance of the dummy control software assessed in simulation.

# STUDY 1: PERFORMANCE IN SIMULATION AND REALITY

In this section, we evaluate `Maple`'s ability to produce control software that crosses the reality gap satisfactorily. To do so, we compare the performance of control software generated by three design methods—`Maple`, `Chocolate` and `EvoStick`—both in simulation and in reality. Previous research (*Francesca et al., 2015*) indicates that `Chocolate` crosses the reality gap more satisfactorily than `EvoStick`. *Francesca et al. (2014, 2015)* argue that `Chocolate`'s ability to cross the reality gap is mainly due to its modular nature. Because `Maple` shares with `Chocolate` the same modular nature and differs from it only in the control architecture adopted, we expect `Maple` to also experience smaller performance drops than `EvoStick`.

We executed each design method 10 times, and thus produced 10 instances of control software. The design budget allocated to each method is 50k simulation runs. The results are depicted in Fig. 4.

### FORAGING

In simulation, the performance of the control software produced by the three automatic design methods is similar, and is significantly better than the one of the dummy strategy. In reality, `EvoStick` is significantly worse than `Maple` and `Chocolate`. The performance of all three methods drops significantly when passing from simulation to reality, but `EvoStick` suffers from the effects of the reality gap the most. See Fig. 4A.

Most of the instances of control software generated by `Maple` and `Chocolate` display similar strategies: the robots explore the environment randomly and once a black area (that is, a source of food) is found, they navigate towards the light to go back to the white

area (that is, the nest). One instance of control software produced by `Maple` uses the *anti-phototaxis* low-level behavior to leave the nest faster once an item has been dropped. Three instances of control software produced by `Chocolate` display an even more sophisticated strategy: the robots only explore the gray area in the search for the sources of food. In other words, the robots always directly leave the nest if they enter it, independently of whether they dropped an item or not.

In simulation, the instances of control software generated by `EvoStick` display drastically different behaviors than the ones produced by `Maple` and `Chocolate`: the robots navigate following circular trajectories that cross at least one food source and the nest. In reality, the robots follow circular trajectories that are much smaller than those displayed in simulation. As a result, the robots tend to cluster near the light. Contrarily to `Maple` and `Chocolate`, and with the exception of few cases, the instances of control software generated by `EvoStick` do not display an effective foraging behavior.

### AGGREGATION

In simulation, `EvoStick` performs significantly better than `Maple` and `Chocolate`, which show similar performance. In reality, we observe an inversion of the ranks: `Maple` and `Chocolate` perform significantly better than `EvoStick`. Indeed, the performance of `EvoStick` drops considerably, whereas the performance drop experienced by `Maple` and `Chocolate` is smaller. See Fig. 4B.

The instances of control software produced by `Maple` and `Chocolate` efficiently search the arena and make the robots stop on the black areas once they are found. In simulation, with the control software produced by `EvoStick`, the robots follow the border of the arena and then adjust their trajectory to converge towards neighboring peers that are already situated on a black spot. In reality, the control software generated by `EvoStick` does not display the same behavior: robots are unable to find the black areas as efficiently as in simulation because they tend to stay close to the borders of the arena. Moreover, the robots tend to leave the black areas quickly when they are found. Although the three design methods perform significantly better than the dummy control software in simulation, none of the methods produced control software that makes the physical robots reach a consensus on the black area on which they should aggregate.

## STUDY 2: PERFORMANCE VERSUS DESIGN BUDGET

In this section, we investigate the performance of `Maple` and `Chocolate` across different design budgets. Because the search space (that is, all instances of control software that can be generated) of `Chocolate` is significantly larger than the one of `Maple`—$O\big(|\mathscr{B}|^4|\mathscr{C}|^{16}\big)$ and $O\big(|\mathscr{B}|^4|\mathscr{C}|^4\big)$, respectively (*Kuckling et al., 2018b*)—we expect `Maple` to converge to high performing solutions faster than `Chocolate`.

We consider 6 design budgets: 0.5k, 1k, 5k, 10k, 50k and 200k simulation runs. For each design budget, we executed each design method 20 times, and thus produced 20 instances of control software. In total, the two design methods have been executed 120 times each. The results are depicted in Fig. 5.

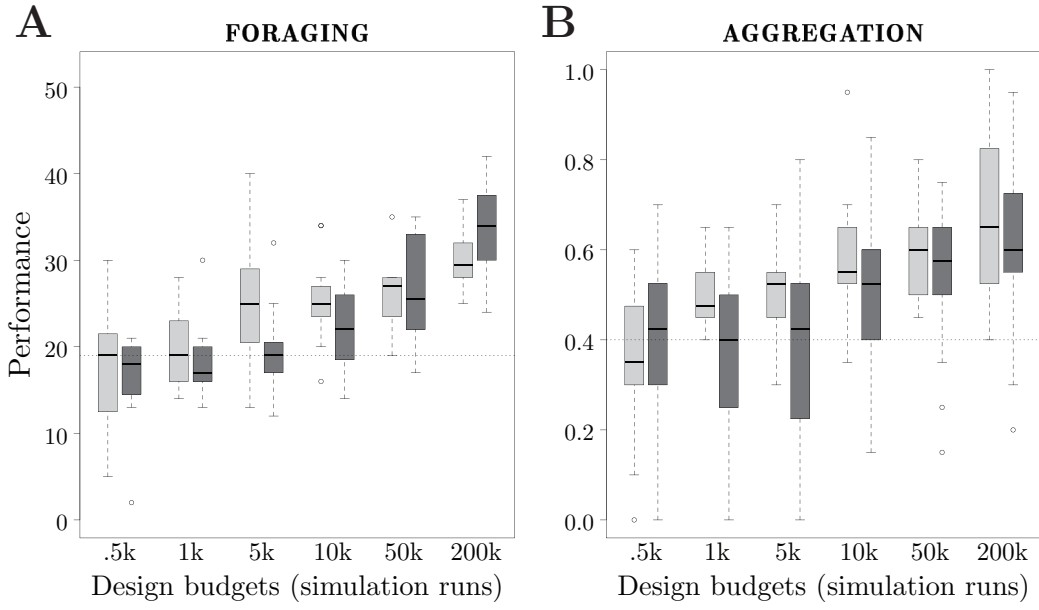

**Figure 5 Results of Study 2.** Performance of `Maple` and `Chocolate` over multiple design budgets, expressed in number of simulation runs. Light gray boxes represent the performance of `Maple`, dark gray boxes the one of `Chocolate`. The dotted line represents the median performance of the dummy control software.

**FORAGING**

The performance of the methods show different trends when the design budget increases. For `Maple`, there is a significant improvement of the performance between design budgets of 1k and 5k, and between 50k and 200k simulation runs. For `Chocolate`, the performance increases significantly between design budgets of 5k and 10k, 10k and 50k, and 50k and 200k simulation runs. See Fig. 5A.

With very small design budgets—0.5k and 1k simulation runs—`Maple` and `Chocolate` show similar performance: they are unable to find solutions that are better than the dummy control software. With a small design budget—5k simulation runs—`Maple` performs significantly better than `Chocolate`. Also, with 5k simulation runs, `Chocolate` and the dummy control software show similar performance. With a large design budget—200k runs— `Chocolate` performs significantly better than `Maple`. Indeed, the instances of control software generated by `Chocolate` display a more sophisticated foraging strategy than those generated by `Maple`: to increase the rate of discovery of the food sources, the robots only explore the gray area of the arena, and stay away from the nest. It appears that, with `Maple`'s restrictions on the structure of the behavior trees, this strategy cannot be produced. Indeed, in the instances of control software that can be produced by `Maple`, only one condition can terminate the execution of the action performed, whereas in the ones produces by `Chocolate`, up to four conditions can. Therefore, with `Maple`, the robots are forced to explore the whole arena until they find the food sources (that is, the black circles). However, it is important to notice that the behavior trees generated by `Maple` with

a design budget of 5k simulation runs are only outperformed by probabilistic finite-state machines when 200k simulation runs are allocated to `Chocolate`.

**AGGREGATION**

The performance of the control software generated by both methods increase almost constantly with the design budget. Also for this mission, `Chocolate` requires a design budget of at least 10k simulation runs in order to generate control software that is significantly better than the dummy control software. Contrarily, `Maple` only requires 1k simulation runs. With 1k and 5k simulation runs, `Maple` outperforms `Chocolate`. For larger design budgets, `Maple` and `Chocolate` show similar performance. See Fig. 5B.

Although the design budgets considered allow the two methods to outperform the dummy control software in multiple occasions, neither of them generated control software that completed the mission satisfactorily. Indeed, the maximal median performance obtained is $F_A = 0.65$, which means that only 13 out of the 20 robots were on the same black spot.

## STUDY 3: `MAPLE` AND SOME OF ITS POSSIBLE VARIANTS

In this section, we explore the changes in performance when variations to the control architecture of `Maple` are introduced. Our exploration is not exhaustive: we only consider variants that generate behavior trees whose structure is similar to the one of the behavior trees generated by `Maple`. We limit our exploration to variants that generate trees with: (i) 3 levels (top-level, inner, and leaf nodes); up to 4 branches connected to the top-level node; and exactly 2 leaf nodes per branch. Many variants are possible, however, because the action nodes of `Maple` can only return *running*, some variants are unable to combine low-level behaviors into meaningful and elaborate individual behaviors. Descriptions of these variants, as well as illustrations, are given as part of Supplemental Material (*Ligot et al., 2020*). In the following, we describe variants that are promising and explain how they behave with the modules considered in `Maple`. We tested the most promising variants by generating control software and evaluating their performance in simulation, and report the results.

### Alternative behavior tree structures

`ICFN` (inverted control-flow nodes): The control-flow nodes are inverted with regard to the ones of `Maple`: the top-level node is a selector[*] and the inner nodes are sequence nodes. See Fig. 6A. In this variant, the action node of a subtree is executed as long as the condition returns *success*, whereas it is executed until the condition returns *success* in `Maple`.

`ND` (negation decorator): A negation decorator node can be instantiated above a condition node. See Fig. 6B. The negation decorator returns *failure* (*success*) if the condition returns *success* (*failure*). With the set of conditions available, it is particularly interesting to place a negation decorator above a condition on the color of the ground perceived (that is *black-*, *gray-*, or *white-floor*). Indeed, placing a negation decorator node above a *neighbor-count* condition is equivalent to having an *inverted-neighbor-count* condition, and vice versa. Similarly, a negation decorator above a *fixed-probability*

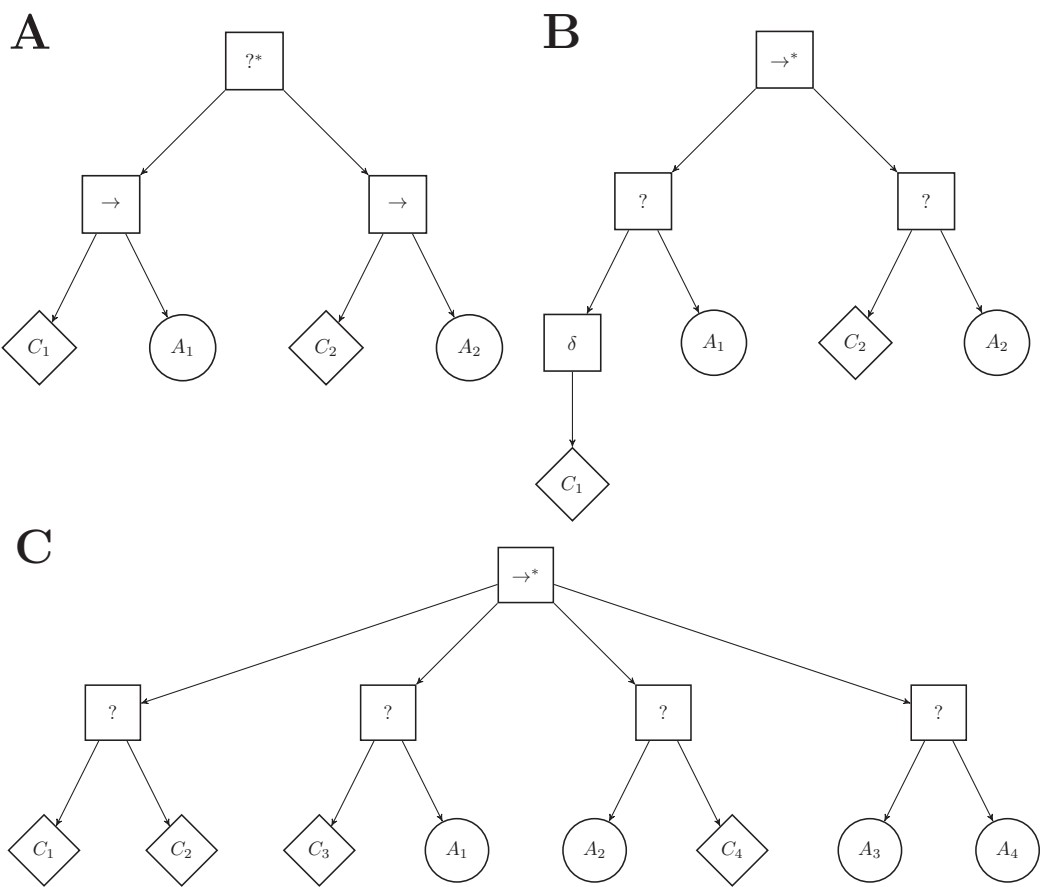

**Figure 6 A few examples of** `Maple`**'s variants.** (A) variant `ICFN` (inverted control flow nodes), (B) variant `ND` (negation decorator), (C) variant `FL` (free leaves). The number of branches connected to the top-level node, and their order, has been chosen arbitrarily.

condition with $\rho$ is equivalent to a *fixed-probability* with $1 - \rho$. However, a negation decorator above a condition on a given color is equivalent to assessing the conditions for the two other colors simultaneously.

`FL` (free leaves): Each leaf node is to be chosen between condition and action nodes. See Fig. 6C. Four pairs of leaf nodes are possible: condition–condition (see first branch), condition–action (which corresponds to the leaf pair imposed in `Maple`, see second branch), action–condition (see third branch), and action–action (see fourth branch). For each subtree, the optimization algorithm is free to chose any pair of leaf nodes. The variant can express disjunction of conditions: a branch following a condition–condition leaf pair is ticked if the first or the second condition is met. However, the variant introduces dead-end states: when an action on the left hand side of a leaf pair is ticked, the action is executed for the remaining of the simulation run.

`CA|CC` (condition–action or condition–condition): The right-hand side leaf node can be a condition or an action node. Two pairs of leaf nodes are thus possible: condition–action, condition–condition. With respect to `FL`, this variant can also express disjunction of conditions, but does not allow for dead-end states.

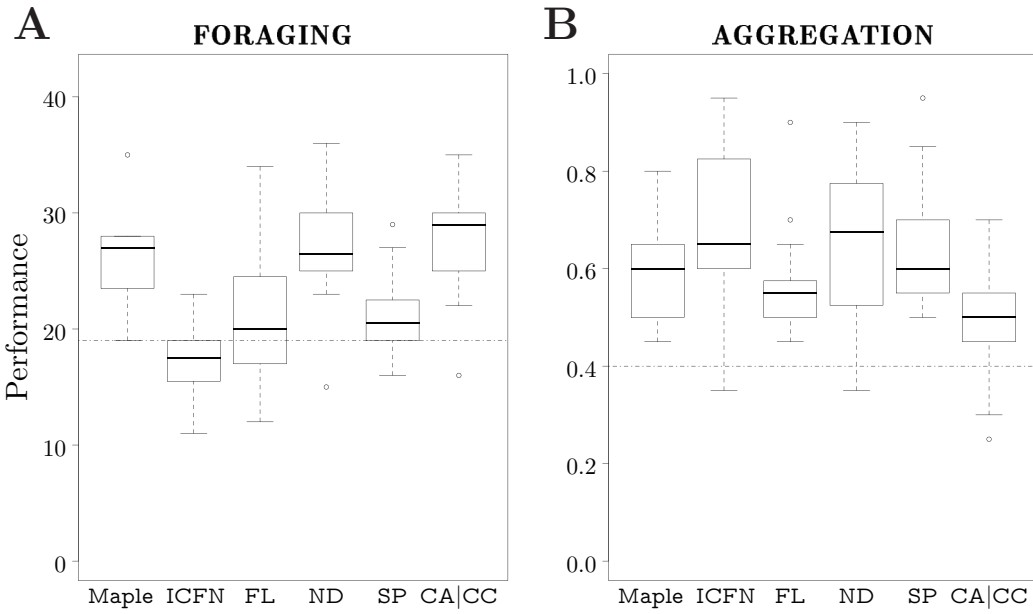

**Figure 7 Results of Study 3.** Performance in simulation of different variants of `Maple`. The dotted line represents the median performance of the dummy control software.

SP (success probability): Each action node has a probability $\rho$ to return *success*. The probability $\rho$ is a real value in the range $[0, 1]$ and is tuned by the optimization process. With this probability, we simulate the capability of the action nodes to assess if the low-level behaviors are successfully executed.

## RESULTS

For each variant, we produced 20 instances of control software, all generated by the same optimization process—Iterated F-race—with a design budget of 50k simulation runs. We compare the performance of the variants to the one of `Maple`.

### FORAGING

None of the variants outperformed `Maple`. `Maple`, `ND`, and `CA|CC` perform similarly; moreover, they outperform `ICFN`, `FL`, `SP`, and the dummy control software. The variants `ICFN`, `FL`, and the dummy control software show similar performance. See Fig. 7A.

All the instances of control software generated by `Maple` show similar behaviors: the robots explore the arena until they find one of the food sources, then navigate towards the nest using the light as a guidance. In some cases, the robots use the *anti-phototaxis* low-level behavior to directly leave the nest once they have deposited an item.

With variant `ND`, we can manually design control software that displays an elaborate strategy: the robots increase the rate at which they discover food sources by only exploring the gray area of the arena. This behavior cannot be expressed by `Maple` (see Study 2). An example of a behavior tree adopting variant `ND` that displays this strategy is illustrated in Section 5 of the Supplemental Material (*Ligot et al., 2020*). In this example, the elaborate

strategy only emerges if the success probability of the condition node below the negation decorator is set to 1. Indeed, if the success probability is slightly lower, the behavior displayed is radically different, and more importantly, inefficient. It appears that, with the allocated budget, this necessary condition makes it unlikely for Iterated F-race to produce this strategy.

Iterated F-race was not able to take advantage of the disjunction of conditions that is available in CA|CC to find better solutions that those of Maple. Indeed, we are unable to do so ourself. However, the increased search space of CA|CC does not hinder the optimization process as results obtained are similar to those of Maple.

In variant SP, the success probabilities, together with the conditions, are termination mechanisms for the subtrees. The additional termination mechanisms makes it harder for Iterated F-race to exploit correlations between conditions and actions that lead to behaviors as efficient as those generated by Maple. Most of the produced control software rely essentially on the *exploration* low-level behavior.

With variant ICFN, one can generate a behavior tree that expresses the same elaborate strategies that can be generated with variant ND (see Section 5 of the Supplemental Material (*Ligot et al., 2020*) for an example). However, ICFN is faced with a similar problem as ND: the success probability of the conditions needs to be set to 1 in order for that elaborate strategy to emerge. With a success probability set to a lower value, the condition node might return *failure* even though its condition is met, and the subtree might therefore terminate prematurely. The allocated design budget was not large enough for Iterated F-race to find behavior trees with meaningful connections between the conditions and behaviors, which resulted in poor performance.

The performance of the variant FL shows the highest variance. Sometimes, the behavior trees generated are similar to those produced by Maple. However, in many cases, the left leaf node of subtrees is an action node with an associated *exploration* low-level behavior. Once this node is reached, this low-level behavior is executed until the end of the experimental run. As a result, the performance observed is similar to the one of the dummy control software.

**AGGREGATION**

Variant ICFN outperforms Maple. Maple, FL, ND, and SP show similar performance. Maple outperforms CA|CC. Every variant produced behavior trees that outperform the dummy control software. See Fig. 7B.

All the instances of control software generated by Maple and the different variants make use of *exploration* and *attraction* as low-level behaviors to efficiently search for black spots. Maple and FL use *stop* as a low-level behavior in order to keep the robots on the discovered spot. Contrarily, the majority of the behavior trees adopting variant ICFN, ND, SP, and CA|CC do not contain the *stop* low-level behaviors as action nodes. Instead, they take advantage of the fact that, when no action node is executed, the robot stands still. ICFN is the only variant for which Iterated F-race was able to exploit this feature to outperform Maple.

## RELATED WORK

Originating from game development (*Isla, 2005*), behavior trees have found recent applications in artificial intelligence (*Perez et al., 2011*) and robotics. Most of the robotics research focuses on their use in manipulators. *Bagnell et al. (2012)* used behavior trees to control a manipulator to perform simple manipulation tasks. *Hu et al. (2015)* used them to control the Raven-II surgical robot to perform an abstract version of tumor ablation surgery. Behavior trees have also been used as a control software for mobile robotics systems. *Marzinotto et al. (2014)* designed a behavior tree to make a NAO robot move towards a table and grasp an object. In all of the presented studies, behavior trees have been manually designed.

To the best of our knowledge, the work of *Jones et al. (2018)* is the first and only application of behavior trees in the context of automatic off-line design of robot swarms[2, 3]. The authors proposed a method based on genetic programing that automatically generates control software for a swarm of Kilobots in the form of behavior trees. The method has been tested on a foraging task in simulation and in reality. Their results suggest that behavior trees can be used as a control architecture in swarm robotics. Besides the different optimization processes adopted in the method of *Jones et al. (2018)* and in `Maple`, another major difference between the two methods lies in the action nodes used. In the method proposed by *Jones et al. (2018)*, low-level behaviors are atomic commands: for example, move forward, turn left/right, or store data. Contrarily, the low-level behaviors that can be combined by `Maple` are more complex actions. Regardless of this difference, the low-level behaviors of the two methods lack natural *success* or *failure* termination criteria. To use their atomic low-level behaviors as action nodes in behavior trees, *Jones et al. (2018)* programed the action nodes such that they return *success* after the second execution of the behavior, but *failure* is never returned. This solution allowed their method to have no restrictions on the selection of the control-flow nodes. In `Maple`, the action nodes can only return *running*, but the structure of the behavior trees, and the control-flow nodes, are restricted such that an external condition terminates the execution of an action.

## CONCLUSIONS

In this article, we presented `Maple`: an automatic modular design method that generates control software for robot swarms in the form of behavior trees. `Maple` is part of the AutoMoDe family: it generates control software by selecting, combining, and fine-tuning a set of predefined modules. Previous instances of AutoMoDe have all used probabilistic finite-state machines as a control architecture. In comparison to finite-state machines, behavior trees offer a number of appealing features. However, most of these features only emerge if the action nodes return their states of execution, that is, if the robot is able to tell whether the low-level behavior it executes is terminated successfully, could not execute normally, or still requires time to terminate. In the context of swarm robotics, the simple and reactive robots typically used are not able to determine the state of execution of the low-level behaviors they operate. With `Maple`, we have investigated the use of behavior trees as a control architecture in the automatic modular design for robot swarms, and have shown that they can still be used even if the low-level behaviors they combine do not

[2] The authors also adapted their system for the onboard evolution of a swarm of nine Xpucks (*Jones et al., 2019*).

[3] *Neupane & Goodrich (2019)* proposed a method based on the grammatical evolution of behavior trees for robot swarms, but their experiments were conducted in simulation only and the focus of the paper is mainly ported on the evolutionary approach rather than on behavior trees.

return *success* nor *failure*. It is our contention that, despite their potential is not exploited in the context of the automatic modular design of robot swarms, behavior trees are a control architecture that is worth exploring. In particular, we reckon that the inherent modularity they offer could be exploited by future automatic modular design methods. In fact, behavior trees can be easily manipulated without compromising their structural integrity, which allows for the use of tailored optimization algorithms based on local manipulations.

We devised `Maple` to be as similar as possible to `Chocolate`: the two methods share the same optimization algorithm, the same set of predefined modules, and generate control software on the basis of the same reference model. The only difference between `Maple` and `Chocolate` is the control architecture adopted. We conducted three studies based on two missions: FORAGING and AGGREGATION to assess the implications of adopting behavior trees as a control architecture. In the first study, we assessed `Maple`'s ability to cross the reality gap satisfactorily by comparing its performance in simulation and in reality against `Chocolate` and `EvoStick`, an evolutionary swarm robotics method. In the second study, we investigated the effect of the design budget on `Maple` and `Chocolate`. In the third study, we explored different variants of `Maple`'s control architecture.

Our main findings are the following. (A) The results show that `Maple` is robust to the reality gap. Indeed, `Maple` and `Chocolate` performed similarly, and they suffered from a reduced performance drop with respect to `EvoStick`. These results confirm *Francesca et al. (2014)* conjecture that AutoMoDe is robust to the reality gap due to its modular nature. They also indicate that the architecture into which the predefined modules are combined is a secondary issue. (B) The study on the effect of the design budget has shown that: (i) the restrictions on the structure of `Maple`'s behavior trees inhibit its expressiveness, indeed, for FORAGING, `Maple` is unable to express some efficient solution that `Chocolate` could generate; (ii) `Maple` converges to efficient solutions faster than `Chocolate` because of the smaller search space. The restrictions of `Maple`, imposed by the absence of natural termination criteria in the low-level behaviors adopted, appear to be a double-edged sword: they facilitate the initial search for efficient solutions, but curb the expressiveness of behavior trees. When adopting the low-level behaviors of `Chocolate`, none of the variants considered outperformed `Maple` in both missions. Overall, our three studies indicate that behavior trees can be used in the particular context of swarm robotics in which low-level behaviors typically do not have a natural termination criterion. However, they also suggest that behavior trees only offer a benefit over probabilistic finite state machines when the design budget is small.

Future work could develop along two avenues. The first one could be dedicated to further investigate the use of `Vanilla`'s and `Chocolate`'s low-level behaviors as action nodes of behavior trees. For example, the control software generated by `Maple` with different design budgets could be assessed in robot experiments. The same holds for control software generated by `Maple`'s variants. Also, further variants could be explored by relaxing the restrictions on the number of levels, branches, and leaves. For the relevant ones, the effect of the design budget could be investigated. As a second avenue, future work could be devoted to developing an ad-hoc optimization algorithm that takes advantage

of the inherent modularity of behavior trees. Local search algorithms, such as iterative improvement and simulated annealing, have shown to be promising algorithms for the automatic modular design of swarm behaviors (*Kuckling, Ubeda Arriaza & Birattari, 2019*; *Kuckling, Stützle & Birattari, 2020*) and could serve as starting points.

### Funding

The project has received funding from the European Research Council (ERC) under the European Union's Horizon 2020 research and innovation program (grant agreement No. 681872). Mauro Birattari and Jonas Kuckling received support from the Belgian Fonds de la Recherche Scientifique—FNRS. The funders had no role in study design, data collection and analysis, decision to publish, or preparation of the manuscript.

### Grant Disclosures

The following grant information was disclosed by the authors:
The project has received funding from the European Research Council (ERC) under the European Union's Horizon 2020 research and innovation program (grant agreement No 681872). Mauro Birattari and Jonas Kuckling received support from the Belgian Fonds de la Recherche Scientifique—FNRS. The funders had no role in study design, data collection and analysis, decision to publish, or preparation of the manuscript.

### Competing Interests

Mauro Birattari is an Academic Editor for PeerJ.

### Author Contributions

- Antoine Ligot conceived and designed the experiments, performed the experiments, analyzed the data, performed the computation work, prepared figures and/or tables, authored or reviewed drafts of the paper, and approved the final draft.
- Jonas Kuckling conceived and designed the experiments, performed the experiments, authored or reviewed drafts of the paper, and approved the final draft.
- Darko Bozhinoski conceived and designed the experiments, authored or reviewed drafts of the paper, and approved the final draft.
- Mauro Birattari conceived and designed the experiments, authored or reviewed drafts of the paper, directed the research, and approved the final draft.

### Data Availability

Data and code are available in the Supplemental Files and at IRIDIA: http://iridia.ulb.ac.be/supp/IridiaSupp2020-009/index.html.

### Supplemental Information

Supplemental information for this article can be found online at http://dx.doi.org/10.7717/peerj-cs.314#supplemental-information.

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
