# Peer review of "Automatic modular design of robot swarms using behavior trees as a control architecture"

_PeerJ Computer Science, doi:10.7717/peerj-cs.314_

## Round 0.1 · original submission · Major Revisions

The authors must thoroughly address all comments and suggestions of the reviewers when revising the manuscript.

·

Basic reporting

This paper concerns the automatic design of controllers for swarm robotics. Building on the AutoMoDe work of Francesca et al, they extend the methodology to encompass the use of behaviour trees as a control architecture. They evaluate the new method, termed Maple, against the Chocolate AutoMoDe variant, and against a neuroevolutionary approach termed Evostick. The evaluation is performed on two classical swarm tasks, aggregation and foraging. They show that, like Chocolate, and unlike Evostick, the new method successfully crosses the reality gap. The paper is generally well written and structured, with good supporting references. Results are presented clearly, and data and code is shared.

Experimental design

The Maple algorithm builds on previous work and reuses a set of basic low-level behaviours and conditions, which corresponded to state machine states and transitions in that work. These behaviours (actions) and conditions are placed as leaf nodes in a restricted behaviour tree structure, consisting of up to four branches, each with exactly two leaf nodes. Iterated F-race is used to optimise the contents of the leaf nodes within this fixed structure. Candidates are evaluated in simulation during the optimisation. Three studies are performed, in each case using two tasks, foraging and aggregation. Study 1 looks at performance in simulation and in reality, hypothesising that the modular nature of Maple should result in similar reality gap crossing behaviour to Chocolate. This is demonstrated with nice results clearly showing the similar advantages of Maple and Chocolate over Evostick. Study 2 compares the performance of Maple and Chocolate at different simulation budgets for optimisation, hypothesising that Maple should converge on high performing solutions faster than Chocolate, since the search space is smaller. This is shown to hold when the simulation budget is small, but Chocolate performs better at higher simulation budgets.

Study 3 looks at the effect of variations of the behaviour tree structure and compares them to the previous Maple results. This section is perhaps overlong, with many variants described then discarded, and the results could be summarised in a boxplot for more clarity. There are interesting hints here about the limits to the Iterated F-race optimiser.

Validity of the findings

I am perfectly happy with the validity of the results.

Additional comments

In Study 1 (line 411) the authors speculate about the cause of the reality gap performance loss with a rather unevidenced assertion about robots unable to move due to frictions. Please can you elaborate on this.

In Study 2, I would like to see some more discussion about the reduced complexity of available solutions limiting ultimate performance of Maple compared to Chocolate.

In Study 3, please shorten the discussion of variants, particularly the uninteresting ones, and summarise comparative results with boxplots.

The argument is advanced that these low-level behaviours have no natural termination criteria and thus do not fit into the standard behaviour tree methodology whereby a node can return success, failure, or running, and can only return running. This is partly used to justify a restricted behaviour tree structure. I think this is not quite correct. The motors could be regarded as contested resource that only a single action node could access in a control step. The first action node to actuate the motors could return a ‘success’, all other action nodes within that control step that are also ticked could not access the motors and would return ‘running’, indicating an uncompleted action. This would allow a more general behaviour tree structure. Perhaps this could be reflected in the future work.

Future work could also expand on possibilities for developing the optimisation algorithm, this is already hinted at.

Minor error: line 103 typo, should be ‘planning’


Overall, I enjoyed this paper and think it is a solid contribution to the field.

Reviewer 2 ·

Basic reporting

The manuscript presents an automatic modular design of robot swarms using behavior trees as a control architecture. Using behavior trees in swarm robotics is a challenging topic. It is possible to find different approaches in literature as well. It is a popular topic for robotics researchers.
The paper is a well-written paper and language is clear.
The abstract should be more specific and deal with the novelties and contribution of this study, strongly.
The authors must emphasize differences between the manuscript and their previous work strongly.
It is necessary to do a new literature review. The introduction part of the manuscript must be revised. The authors should improve the introduction part according to a new literature review.
Figures and tables are good enough.

Experimental design

Manuscript's title and abstract are the same with supplementary files of the authors' previous work which was published at the ANTS 2018 conference in 2018. The necessary links are listed below. The authors must emphasize differences between the manuscript and their previous work strongly. And, in my opinion, they must change the abstract and title of the manuscript.

- The abstract should be more specific and deal with the novelties and contribution of this study, strongly. And authors must emphasize differences between the manuscript and their previous work strongly.
- Experimental studies are good enough.
- The manuscript contains significant outcomes.

Validity of the findings

- Experimental studies are good enough.
- The manuscript contains significant outcomes.

Additional comments

First of all, I would like to thank for authors' valuable work. The manuscript presents an automatic modular design of robot swarms using behavior trees as a control architecture. Using behavior trees in swarm robotics is a challenging topic. It is possible to find different approaches in literature as well. It is a popular topic for robotics researchers. The paper is a well-written paper and its topic is important. I have a few suggestions to authors which are listed below.

- The abstract should be more specific and deal with the novelties and contribution of this study, strongly. And authors must emphasize differences between the manuscript and their previous work strongly.
- It is necessary to do a new literature review. The introduction part of the manuscript must be revised. The authors should improve the introduction part according to a new literature review.

---

## Round 0.2 · accepted · Accept

The revisions done are sufficient for the manuscript to be accepted.

·

Basic reporting

No comment

Experimental design

No comment

Validity of the findings

No comment

Additional comments

I am happy that the authors have addressed the points raised in my previous review.